# Quality of Life and Health among People Living in an Industrial Area of Poland

**DOI:** 10.3390/ijerph16071221

**Published:** 2019-04-05

**Authors:** Szymon Szemik, Małgorzata Kowalska, Halina Kulik

**Affiliations:** 1Department of Nursing Propaedeutics, School of Health Sciences, Medical University of Silesia in Katowice, 20/24 Francuska Street, 40-027 Katowice, Poland; hbkulik@gmail.com; 2Department of Epidemiology, School of Medicine, Medical University of Silesia in Katowice, 18 Medyków Street, 40-027 Katowice, Poland; mkowalska@sum.edu.pl

**Keywords:** quality of life, health status, employment, cross-sectional study

## Abstract

*Background:* The quality of life and health status of the population significantly depends on socio-economic factors, including working and employment conditions. *Methods:* This epidemiological cross-sectional study was carried out among young inhabitants aged 25–44 years living in the Silesian voivodeship in Poland. The quality of life was evaluated using the short version of the WHOQOL-BREF questionnaire. *Results:* A total of 905 respondents were examined. It was shown that the poor quality of life in all assessed domains was associated with a low job satisfaction level, low physical activity, and higher self-assessed health status. Furthermore, the worse self-assessed health status in the study group was mainly related to such factors as earlier diagnosed chronic disease, lower job satisfaction, and low physical activity. Additionally, diagnosed chronic disease among occupationally active respondents was correlated with health deterioration due to excessive stress, living in the vicinity of heavy road traffic, and was declared by women more frequently. *Conclusions:* The results of the presented study confirmed that the quality of life and health status in young inhabitants of the Silesian voivodeship significantly depends on the work characteristics, employment conditions and lifestyle factors.

## 1. Introduction

Quality of life (QoL) is a cross-disciplinary concept, and this issue is described in the research of social science as well as in medical and health sciences. The assessment of quality of life is usually based on multidimensional procedures, including macroeconomic indicators (GDP—gross domestic product, unemployment rate), health (suicide rate, mortality rate, access to health services), education (GER—gross enrolment ratio), and many subjective factors such as individual sense of happiness, family relationships, and sense of community [1]. Published data available indicates that the level of quality of life is strongly associated with psychosocial working conditions [2,3,4], and is also related to the type of work [5]. Indeed, affirmative occupational activity can positively affect the quality of life and health, and it has a positive impact on general mental well-being and prevent depression [6,7]. On the other hand, some studies have confirmed the negative influence of unemployment on life satisfaction and self-assessed health status [8,9]. Furthermore, specific elements of work such as rotating shift work correlate with a decreased quality of life and are associated with an increase in coronary heart disease risk and independently contributes to weight gain [9,10]. Moreover, occupational stress leads to some problems, for instance, chronic fatigue syndrome or burnout [11,12,13].

The research presented in this paper was conducted in a population of The Silesian voivodeship located in southern Poland, the most industrialized region of the country. The dominant industries are mining, iron, lead and zinc metallurgy, engineering, automobile, chemical, building materials, and textile. It is worth indicating that, according to the Local Data Bank of Statistics Poland, the percentage of workers employed in industry and exposed to health risk factors is the highest in the whole of Poland. Furthermore, the percentage of the population aged 25–44 remains one of the highest, at 27.5%. Therefore, we believe that the above arguments justify our decision to study the problem of quality of life in this sector of the population.

### Objective

The objective of the presented paper is to identify and evaluate determinants of quality of life level and self-assessed health status, and to assess the relationship between significant factors and health and quality of life in young adults who are occupationally active inhabitants of The Silesian voivodeship.

## 2. Materials and Methods

### 2.1. Study Design and Sampling

The presented QoL research is based on the data obtained in an epidemiological cross-sectional study among young inhabitants of the Silesian voivodeship (an industrial region in southern Poland). According to the current Local Data Bank of Statistics Poland (2016), the total population of the Silesian voivodeship is 5,073,170 (Table 1), with an overrepresentation of women (52.89%). Furthermore, the value of femininity ratio shows that among the general population, there are 107 women for every 100 men. Moreover, the value of age dependency ratio indicates that there are 61.8 dependents (people below 15 and over 65 years) for every hundred people of working age (15–64 years). Finally, it was shown that the employment rate in the general population was higher among men (58.4% vs. 44.4% among women), and this observation is typical for the entire country. The demographic situation suggests temporal differences between the years 2000 and 2016. It is worth noting that the population of the Silesian voivodeship is growing older. The number of people aged 65 doubled during this period.

Therefore, the research group was selected from the general population of 1,392,913 inhabitants of the Silesian voivodeship aged 25–44 years. The sample size of this study was calculated based on the OpenEpi tool. The suggested minimum number of participants amounted to approx. 600–2000, depending on the accepted values of confidence interval and confidence limit. 2000 participants were randomly selected from within the address database obtained from the Polish Resident Identification Number Register (PESEL). Personal invitations with the questionnaire and reply paid envelope were sent for each one of the respondents during a period from November 2016 to March 2017. Inclusion criteria were: the place of residence in the Silesian voivodeship, age between 25 and 44 years, and the provision of informed consent. As 905 completed questionnaires were received, the response rate was 30.2%. The ethics approval for the study was received from the Bioethical Committee of the Medical University of Silesia in Katowice (approval number KNW/0022/KB/48/15; date: 17.03.2015). Informed consent was obtained from all participants.

### 2.2. Quality of Life Measurement

The quality of life of respondents was evaluated using the Polish version of the WHOQOL-BREF questionnaire. The Polish version of this questionnaire is available at the World Health Organization (WHO) website and is a reliable tool for assessing the quality of life among economically active adults in Poland [14]. The questionnaire contains 26 items as factors of QoL in four domains: Somatic (physical health), Psychological, Social (social relationships) and Environmental. Respondents were asked via questionnaire how they perceive each aspect of quality of life to assess them either as satisfactory or problematic. The somatic domain includes energy and fatigue, mobility, pain and discomfort, the need for medical treatment, sleep and rest, as well as satisfaction with the capacity of work. The psychological domain, among others, contains satisfaction with body, appearance and the frequency of negative and positive feelings in daily life. The social relationships domain presented questions regarding satisfaction with personal relationships, social support system, and sexual activity. The last environment domain includes questions pertaining to physical safety and security, availability of financial resources, accessibility, and quality of social care and health services, home and physical environment satisfaction, opportunities for leisure time activities, and also transport. Under the recommendation of the WHO, the raw values were transformed into a scale from 0 to 100 points.

Moreover, the final version of the questionnaire used also contains a dozen authorial questions identifying socio-demographic data (age, sex, marital status, education level and place of residence), lifestyle (physical activity, drinking alcohol, smoking, living in the vicinity of heavy road traffic), health (chronic diseases, self-assessed health status) and factors related to occupational activity (type of job, amount of income, job satisfaction level and impact of workplace on health deterioration). It is important to explain that the questions about the type of job include categories such as a white collar (mental work) and blue collar (physical work). Moreover, the question about income relates to the amount of average disposable income in The Silesian voivodeship (criterion 1318 PLN) according to the data from the Local Data Bank of Statistics Poland (data from 2016).

### 2.3. Statistical Analysis

Statistical analysis was undertaken using the Statistica 12.0 package (Dell Software, Inc., Round Rock, TX, USA). The missing values limiting aggregated evaluation of the quality of life were removed from further statistical analysis. The measures of central tendency (median, quartile) and dispersion (IQR—interquartile range) were applied in the statistical description of QoL domains. Then, the qualitative variables were presented by frequency and percentage. Differences between groups were tested by U Mann–Whitney or Kruskal–Wallis test for the continuous variables and chi-squared test for categorical variables, respectively. In all analyses, *p* values below 0.05 value were considered as statistically significant. Finally, the relevant relationships between particular variables were verified in multivariable analysis. Only the statistically significant variables obtained in bivariate analyses were included in the multivariable or logistic regression models. Both regression models were developed using backward eliminations of the statistically insignificant predictors. Additionally, the Hosmer–Lemeshow test was used to evaluate the goodness of fit of logistic regression models, *p* value above 0.05 indicates a good fit of the model.

## 3. Results

The examined sample comprised 905 inhabitants of the Silesian voivodeship aged 25–44 years old. Sociodemographic variables are shown in Table 2. An overrepresentation of women and younger respondents (aged 25–34 years old) was observed in the study group. On the other hand, the percentage of men and older people in the whole research population was lower (43.87% men and 45.31% people aged 35–44 years old). Additionally, the respondents mostly declared as having a higher education level (63.65%), and more participants were in a relationship (67.18%). The majority of the study group declared good self-assessed health status (60.22%) and a higher quality of life (76.13%). Furthermore, 21.44% of the respondents declared chronic disease as diagnosed by a medical doctor.

Table 3 shows the self-declared characteristics of the occupational environment in the study group. Over half of respondents represented blue collar workers (54.48%); furthermore, the majority of them declared the amount of monthly income per person in the household of 1318 PLN or more. 622 people (68.72%) indicated a higher level of job satisfaction. 38.34% of the participants manifested a deterioration in health due to their working conditions, mostly as a result of excessive stress (23.87%) and physical strain (18.12%).

The selected descriptive statistics that identify the scoring of particular QoL domains in the study group are shown in Table 4. The highest scores were attributed to the social domain of quality of life, whereas the environmental domain was the lowest. It is worth underlining that each assessed domain of QoL attained more than half of the total scoring.

In the next step, the total scoring of a particular domain of QoL was analyzed in groups defined by independent variables using nonparametric tests (U Mann–Whitney or Kruskal–Wallis test, respectively) and was verified in the multivariable regression model (Table 5). It was shown that the poor QoL in all assessed domains was associated with low job satisfaction levels, low physical activity, and higher self-assessed health status. Moreover, worse quality of life in the field of the somatic and environmental domains was related to low monthly income and low education level.

Additionally, the relevant relations between self-assessed health status and particular independent variables were tested in a logistic regression model. Results are demonstrated in Figure 1. Worse self-assessed health status in the study group was mainly related to such factors as a diagnosed chronic disease (OR = 3.34), lower job satisfaction (OR = 2.43), and low physical activity (OR = 2.32). Moreover, poor self-assessed health status was also significantly associated with performing work that deteriorates health condition (OR = 1.72), even more than smoking tobacco (OR = 1.60) and younger age (OR = 1.58).

Meanwhile, a diagnosed chronic disease (Figure 2) among occupationally active respondents was correlated with health deterioration due to excessive stress (OR = 2.05) and living in the vicinity of heavy road traffic, and was more frequently declared by women (OR = 1.49).

## 4. Discussion

The results obtained confirmed that job satisfaction level and frequency of physical activity were the most important determinants of the quality of life and self-reported health status in occupationally active inhabitants of the Silesian voivodeship aged 25–44 years old. In addition, the quality of life in the study group correlated with the self-reported health status, education level and the level of income.

However, it should be noted that the current study group is characterized by the overrepresentation of women, younger respondents, and people declaring a higher level of education. This phenomenon is probably related to a greater tendency for female, younger and better-educated respondents to participate in questionnaire surveys [15]. Moreover, based on Finnish research, females are more interested in and reported much more seeking of health-related information and are more attentive to their health than men [16]. Additionally, higher education level is strongly associated with positive health behaviors, such as preventive healthcare [17]. Because of these facts, we suppose that females and higher educated people are more likely to respond to our health and quality of life survey.

A lack of other published data on QoL among young employees in Poland justifies the necessity of comparing results with the previous observations undertaken in 2010 in a population of older inhabitants of the Silesian voivodeship aged 45–60 years old [18]. It was recognized that the average scoring of all assessed quality of life domains was higher among younger respondents (25–44 years). The largest difference concerned the somatic domain of quality of life (Me = 64.29 vs. Me = 56.00 among people aged 25–44 and 45–60 years old, respectively). Additionally, the level of QoL among both groups was associated with a higher level of education and better health status. What is more, a good self-assessed health status among both groups was determined by safer work (without exposure to risk factors) and without an unhealthy lifestyle. Last but not least, a good health status among the older group was also related to having social relations (partner or spouse) and additionally to not living in the vicinity of heavy road traffic. A better quality of life is strongly related with younger age, but we couldn’t exclude that the observed differences may have been related with the different socio-economic situation between the years 2010 and 2016 in the same study region. Current data confirm the improve of the economic situation of inhabitants in the Silesian voivodeship.

What is more, obtained results confirmed that age is a significant factor of psychological quality of life domain. Younger respondents (aged 25–34 years) declared a better quality of life than somewhat older (aged 35–44 years). Due to potential collinearity between age, quality of life and other sociodemographic determinants, additional statistical modelling was conducted. It was ascertained that younger people were more frequently people with a higher education level, a higher level of income and better self-assessed health status. Moreover, younger respondents rarely declared having been diagnosed with the chronic disease or health deterioration due to working conditions. All of these factors significantly affect the better quality of life in the study group.

Numerous studies regarding the influence of occupational activity on quality of life and/or health status in young adults raise the problem of unemployment. It has been documented that unemployment had a negative impact on the QoL and health status among people among the younger productive age. A longitudinal study among young Swedish people confirmed that long-term unemployment contributed to a deterioration of health status in women and increase alcohol intake among men [19]. Moreover, based on German and Austrian research, unemployment has a negative impact on life satisfaction and on declared mental health among the younger population [8,9]. Simultaneously, there is evidence that re-employment leads to a significant improvement in self-assessed health status and quality of life, especially in young adults. For instance, a Chinese study showed that the level of quality of life is higher among occupationally active respondents in comparison to unemployed people [20].

It needs to be underlined that the influence of occupational activity on QoL and health status depends on occupational risk factors. The results obtained in this study confirmed that a worse quality of life in the environmental domain and more frequently declared chronic disease significantly concerned people exposed to excessive stress at work. A similar connection is very often indicated in much of the research available: it has been shown that a high workload is associated with higher risk of coronary artery disease [21], obesity [22,23], arterial hypertension [24] and depression [25]. Furthermore, high job strain increases the risk of long-term sickness absence [25], chronic fatigue syndrome [11] and burnout [12,26].

The observation about the positive influence of job satisfaction on the quality of life and health status in own study is compatible with the results of other scientists’ results. It has been indicated that the higher job satisfaction and lower level of job stress are significantly related to the higher score of QoL among hospital staff [26]. Moreover, there is evidence that increased job satisfaction level is related to increased physical and mental health [27]. It is very important to explain what aspects of work have an impact on job satisfaction level and as a result of the quality of life and health. Based on the results of the cross-sectional study presented, it is not possible to explain what exact aspects of work are significantly associated with job satisfaction. Those limitations highly justify a recommendation to conduct in-depth research in young Polish employees. Physical activity is the next significant factor improving the QoL and self-assessed health status of the young adults in the presented study.

A positive relation between quality of life and general health has been recognized in numerous publications. What is very important, here, is that taking up physical activity improves the quality of life in people aged 25–50 years old [28]. Likewise, physical activity is also beneficial for the physical and psychosocial health of children and youth [29,30,31]. Also in the case of elderly people, physical activity is associated with better quality of life, furthermore it prevents chronic diseases and supports their independence [32]. Finally, physical activity is a part of therapeutic rehabilitation in many diseases, including cancer and depression [33,34,35].

The observations revealed are important for developing and improving the effectiveness of prevention programs implemented in relation to professionally active employees. The progressive aging of the population and the accompanying increase in the number of chronic diseases are forcing individuals responsible for the health of employees to participate in the preventive activity. As has been shown, QoL and health status improvement can be achieved, among others by preventing excessive workload, chronic stress, and by supporting employees in improving their physical activity.

### Limitations of the Study

The authors realize that despite efforts to eliminate selection bias, overrepresentation of younger and well-educated people, as well as women, was obtained. The results attained in the cross-sectional study in which the response rate was low (30.2%) should be treated with due caution. Nonetheless, it is worth indicating that distribution of key factors characterizing research samples such as sex and age were compared with the source population data.

## 5. Conclusions

The results confirmed that the level of the quality of life and health status in young inhabitants of the Silesian voivodeship significantly depends on the work characteristics and employment conditions.Job satisfaction level, the level of income and occupation without exposure to excessive stress are significant determinants of better quality of life and health.The level of QoL and health status among the study group is significantly connected with lifestyle factors, especially with physical activity.

## Figures and Tables

**Figure 1 ijerph-16-01221-f001:**
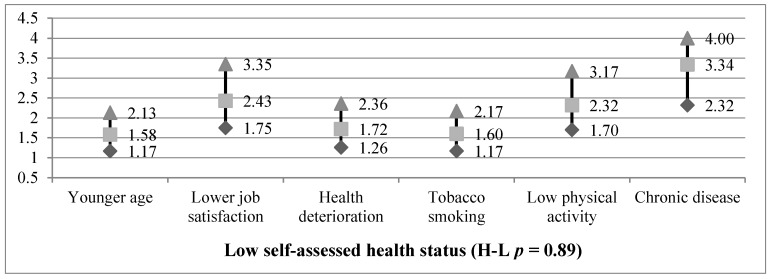
Odds ratio and its 95% confidence interval for low self-assessed health status and particular independent variables (*N* = 905). H-L—Hosmer–Lemeshow test; *p*—statistical significance.

**Figure 2 ijerph-16-01221-f002:**
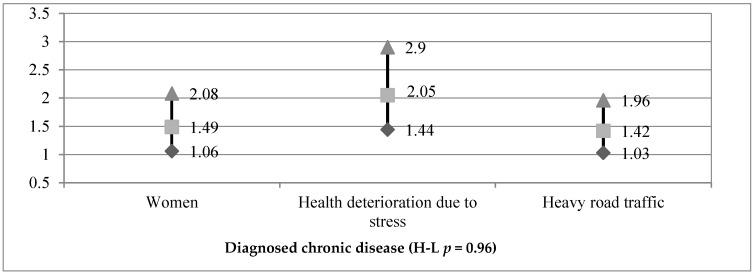
Odds ratio and 95% confidence interval for diagnosed chronic disease and particular independent variables (*N* = 905).

**Table 1 ijerph-16-01221-t001:** Distribution of selected demographic characteristics in the population of the Silesian voivodeship.

Year	2016		2000	
Gender	Women	Men	Women	Men
Number of residents by age				
less than 15 years	314,845	330,597	407,180	428,019
15–24	225,516	235,444	391,894	401,318
25–44	689,159	703,754	688,683	694,470
45–64	658,732	613,540	622,212	577,032
aged 65 and over	794,726	506,857	343,019	205,117
**Femininity ratio**	107		106	
**Total dependency ratio**	61.8		41.0	
**Employment rate**	44.4%	34.8%	34.8%	50.1%
**Unemployment rate**	5.5%	21.0%	21.0%	14.6%

**Table 2 ijerph-16-01221-t002:** Selected descriptive statistics for the study group (*N* = 905).

Sociodemographic Variables		*N*	%
Sex	Women	508	56.13
	Men	397	43.87
Age	25–34 years	495	54.69
	35–44 years	410	45.31
Marital status	in relationship	608	67.18
	single	297	32.82
Education level	higher	576	63.65
	lower	329	36.35
Current tobacco smoking	yes	167	18.45
	no	738	81.45
Ever tobacco smoking	yes	355	39.23
	no	550	60.77
Frequency of drinking alcohol	high	485	53.60
	low	420	46.40
Frequency of physical activity	high	354	39.12
	low	551	60.88
Living in the vicinity of heavy road traffic	yes	424	46.85
	no	481	53.15
Diagnosed by physician chronic disease	yes	194	21.44
	no	711	78.56
Self-assessed health status	good	545	60.22
	bad	360	39.78
Declared quality of life	high	689	76.13
	low	216	23.87

**Table 3 ijerph-16-01221-t003:** Self-declared characteristics of the occupational environment of the respondents (*N* = 905).

Variable		*N*	%
Type of occupational activity	white collar	412	45.52
	blue collar	493	54.48
Degree of job satisfaction	higher	622	68.72
	lower	283	31.27
The level of income	higher (more than 1318 PLN)	658	72.70
	lower (below 1318 PLN)	247	27.30
Impact of the workplace on health deterioration	yes	347	38.34
	no	558	61.66
Health deterioration as a result of excessive stress	yes	216	23.87
	no	689	76.13
Health deterioration due to physical strain	yes	164	18.12
	no	741	81.88
Dull and repetitive job	yes	62	6.85
	no	843	93.15
Excessive noise	yes	69	7.62
	no	836	92.38
Exposure to dust/smell	yes	64	7.07
	no	841	92.93
Uncomfortable temperature	yes	68	7.51
	no	837	92.49
Shift work	yes	69	7.62
	no	836	92.38

**Table 4 ijerph-16-01221-t004:** Summary of the WHOQOL-BREF domains (score after transformation to the scale of 0–100), (*N* = 905).

Quality of Life Domain	Me (IQR)	Range	Q_1_	Q_3_
Somatic	64.29 (25.00)	92.86	53.57	78.57
Psychological	66.67 (16.67)	83.33	58.33	75.00
Social relationships	75.00 (25.00)	100.00	58.33	83.33
Environmental	62.50 (15.63)	87.50	53.13	68.75

Me, median; IQR, interquartile range; Q_1_, first quartile; Q_3_, third quartile.

**Table 5 ijerph-16-01221-t005:** Results of multivariable analysis for the relation between QoL domains and particular independent variables (*N* = 905).

Independent Variable	Regression Coefficient (95%CI ^a^)	*p*-Value ^b^
Somatic (R^2 c^ = 0.22, *p* = 0.00 ^d^)		
Education level (1 = higher, 2 = lower)	0.07 (0.01, 0.13)	0.011
Degree of job satisfaction (1 = higher, 2 = lower)	0.20 (0.14, 0.26)	<0.001
Income (1 = more than 1318 PLN, 2 = below 1318 PLN)	0.11 (0.05, 0.17)	<0.001
Self-assessed health status (1 = good, 2 bad)	0.27 (0.21, 0.33)	<0.001
Frequency of physical activity (1 = high, 2 = low)	0.13 (0.07, 0.19)	<0.001
The place of residence (1 = small town or village, 2 = big city	0.09 (0.03, 0.15)	0.001
Psychological (R^2 c^ = 0.24, *p* = 0.00 ^d^)		
Age (1 = 25–34 years, 2 = 35–44 years)	0.09 (0.03, 0.15)	0.001
Education level (1 = higher, 2 = lower)	0.12 (0.06, 0.18)	<0.001
Degree of job satisfaction (1 = higher, 2 = lower)	0.21 (0.15, 0.27)	<0.001
Self-assessed health status (1 = good, 2 = bad)	0.30 (0.24, 0.36)	<0.001
Frequency of physical activity (1 = high, 2 = low)	0.06 (0.008, 0.12)	0.002
Social Relationships (R^2 c^ = 0.20, *p* = 0.00 ^d^)		
Degree of job satisfaction (1 = higher, 2 = lower)	0.23 (0.17, 0.29)	<0.001
Self-assessed health status (1 = good, 2 = bad)	0.26 (0.20–0.32)	<0.001
Diagnosed chronic disease (1 = yes, 2 = no)	−0.09 (−0.15, −0.03)	0.001
Frequency of physical activity (1 = high, 2 = low)	0.09 (0.03-0.15)	0.001
Environmental (R^2 c^ = 0.28, *p* = 0.00 ^d^)		
Education level (1 = higher, 2 = lower)	0.12 (0.06, 0.18)	<0.001
Degree of job satisfaction (1 = higher, 2 = lower)	0.20 (0.15, 0.26)	<0.001
Income (1 = more than 1318 PLN, 2 = below 1318 PLN)	0.15 (0.10, 0.21)	<0.001
Health deterioration as a results of excessive stress (1 = yes, 2 = no)	−0.14 (−0.20, −0.08)	<0.001
Self-assessed health status (1 = good, 2 = bad)	0.25 (0.19, 0.31)	<0.001
Frequency of physical activity (1 = high, 2 = low)	0.08 (0.02, 0.14)	0.004

^a^ CI, Confidence Interval. ^b^
*p* Value, significance in relation to the reference group. ^c^ R^2^, determination of the model. ^d^
*p*, the significance of the multivariable regression model.

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
