# Peer review of "Quality of Life and Health among People Living in an Industrial Area of Poland"

_ijerph, 2019, doi:10.3390/ijerph16071221_

Round 1

Reviewer 1 Report

Sir, 

first at all thank you for the opportunity to review this very interesting paper. Studies on the QoL of working age population are of significant interest for Public Health, as such researches have the potential to share some lights on the complex interplay between occupational and personal / individual factors on the perceived quality of life / health status. More specifically, Authors suggest that in the Polish population they evaluated (i.e. 25-44 yrs old Silesian residents) work characteristics and employment conditions are as significant as lifestyle factors such as physical activity, eduational status, etc. Even though such results "will not shatter the Earth", they are sounds, the methods are appropriate to the aims. In other words, I think that the present study may deserve a full publication in IJERPH.

However, some significant improvements are required.

First and foremost: why studying such factors in the Silesian youth may be of some interests for international readers? Are the occupational settings / employment conditions of Silesian region characterized by some specificities?  Moreover: please be aware that most international readers are not accostumed to the reality of Poland. So, please include in section 2.1 some more details on the Silesian general population, i.e. employment rates, total number of residents, demography, etc. Again, it seems (Ref. 15) that around a decade ago you have performed a similar study on an older occupational group, including the "older workers". A more extensive confrontation between the two studies would be of significant interest for the international readers, and would shed some more lights on the specificies of the study you have more recently performed. Obviously, please explain the economic and demographic differences between the Silesian Poland of the late 2000's and the same region in late 2010's.

Second, Authors identfied age as a significant factor in the modelling of psychological factors. As they focused their study on a specific subgroup of the general Silesian poulation, and this subgroup was specifically age-defined, this factors should be more extensively presented in the discussion section. In facts, some colinearity inbetween age, education level, work conditions and eventually employment rates should be preventively assumed when studying this specific age group (i.e. I would expect in the age group 35-44 a higher share of participant with higher educational level, and having a more stable and somehow satisfying working condition), and should be therefore extensively treated among limitations or at least discussed. This is an issue even more significant to address as you otherwise report that "the respondents mostly declared to have a higher education level", I suspect you're refering to the general population. Again, this factor may have affected also "per se" your results (well educated people are more likely to report health issues or health complaints).

Third, please explain in deeper details the "effort" (see row 208) you performed in order to obtain a more appropriate representation of the general Silesian population.In fact, the over-representation of the female sex and younger subjects deserves some more discussion, and may found some explanations in the literacy of participants (there are actual differences inbetween 35-44 years and 25-34 years?) as well as in the well-known sensibility of female workers towards their respective health issues. Again, please discuss such factors.

Fourth, at row 58 you report that you used "the Polish version of WHOQOL-BREF questionnaire". This a validated and sound questionnaire, but please include the reference(s) to the validation of its translation (in case of a questionnaire developed by some Health Authorities, please include the specific reference) OR (in case of a self-made Polish translation) please specify. 

Finally, some more suggestions. Firstly, please be aware that the definition of "work related stress" is not consistent even across European Union. Therefore, please include your working definition.

Second, in the section 2.3 you reported an inconsistent formal (not substantial) reference to the p value, initially (row 88), it is described as "P value below 5%", then (row 94) "P value above 0.05". Obviously, the significance level is the same, but choice how to present it to the reader and stick with it (either 5% or 0.05).

Finally, there are some minor phrasing uncertainties scattered across the text and very few typos (e.g. row 56 "30,2%" instead of "30.2%"). Please fix them with your next version.

Best regards.

Author Response

Dear Sir/Madamme,

We really appreciate the effort given during reviewing and insightful reading of our manuscript. We tried to incorporate all of the excellent comments into the text and answer the reviewer’s questions to explain any misunderstandings. We believe that the changes given in the reviewed version of the manuscript are adequate. Please kindly see our responses in atteched files.

Best regards,

Szymon Szemik

Reviewer 2 Report

I congratulate the authors on an ambitious observational research. The investigation is robust and the design well considered. I look forward to seeing the end result of this study when it is finally complete and published. I commend the authors for their work - both all of the work leading up to this point and for the planning of this study - their contribution to the occupationally active people living related with the quality of life and health literature. I do have some comments about certain methodological issues covered below under MAJOR ISSUES:

TITLE

The title of this manuscript are a little long. Perhaps a more concise version for clarity, interes and ease of read.

ABSTRACT

It is hard to get the detail in an abstract when the word count is limited and this is often the hardest part of a paper to write. However, I do feel that it would be beneficial to explain what specifically you are looking at in relation to the occupationally active people living related related with the quality of life and health (this also applies to the main body of the paper). Is it the development of reasons structural for

the occupationally active people living related with QoL.  This needs to be made clearer throughout the paper.

KEYWORDS:

Please use recognised MeSH terms as this will assist others when they are searching for information on your research topic. The following website will provide these (simply start typing in a keyword and see if it exists or find an alternative if it does not): https://www.ncbi.nlm.nih.gov/mesh

INTRODUCTION: 
The introduction is poor, needs to present a better rationale for the study and the methodology employed. Also, the research question itself is sound, however the topic is not strongly introduced. No clear explanation is given to what constitutes health problems in Silesian voivodeship, is it based on  all people in POland ?. Additional information and prevalence would benefit the reader (and several further sources are available prevalence is higher in  females or in males , but only one source. Overall, the introduction would benefit from a broader literature review and more detail on what the problem is, how much it impacts on occupationally active people living related with the quality of life and health literature implications of this would add depth to the intro. Further, to include the work hypothesis in this section

MATERIALS AND METHODS: 

There is no information about validation of the WHOQOL-BREF questionnaire included in the analysis including some explanation about the domains Also, include description of this questionnaire, its reliability and validity and the actual measurements.

Thus, neither appear information related with inclusion and exclusion criteria.  The study design is a cross-sectional study of ramdom sampling method, where the study was conducted in the hospital or in the outpatient center?.

Please describe the ethical and legal and include the date and code register number of ethics committee

Lastly, please, expand and clarification information related with the calculate sample size.

RESULTS: 
The statistical analysis is very simple, I would been liked to see the matrix of correlation or the initial factor matrix 

DISCUSSION: 
I am struggling to make sense of some of this, I am afraid it needs extensive revision. What are the clinical and non clinical implications of your study? How this will inform future larger studies?

CONCLUSION:

These conclusions need to be softened, modified a in order to reflect only the study findings.

Author Response

Dear Sir/Madame,

We really appreciate the effort given during reviewing and insightful reading of our manuscript. We have tried to incorporate all of the excellent comments into the text and answer the reviewer’s questions to explain any misunderstandings. We believe that the changes given in the reviewed version of the manuscript are adequate. Please kindly see our responses in attached file.

Best regards,

Szymon Szemik

Round 2

Reviewer 2 Report

The authors have satisfactorily responded to all of my comments.

Author Response

Dear Sir/Madame,

Thank you for your kind words. We really appreciate the effort given during the second review of our manuscript.

Best regards,

Szymon Szemik